# Advances and Gaps in Natech Quantitative Risk Analysis

**Adriana Mesa-Gómez [1,2,\*], Joaquim Casal [2], Mauricio Sánchez-Silva [1] and Felipe Muñoz [3]**

[1] Civil and Environmental Engineering Department, Universidad de los Andes, Cra 1 N, 18A-10, Bogotá 111711, Colombia; msanchez@uniandes.edu.co

[2] Centre for Technological Risk Studies (CERTEC), Barcelona East School of Engineering (EEBE), Universitat Politècnica de Catalunya, BarcelonaTech (UPC), 08019 Barcelona, Catalonia, Spain; joaquim.casal@upc.edu

[3] Empresa Colombiana de Petróleos (ECOPETROL) S.A., Cra 7 No. 32-42, Bogotá 111711, Colombia; fmunoz@uniandes.edu.co

\* Correspondence: adriana.maria.mesa@upc.edu; Tel.: +57-312-375-3353

**Abstract:** The occurrence of Natech (natural hazard triggering technological disasters) accidents has generated a reflection about the need to manage adequately the risk to people, to the environment, and to the infrastructures subjected to natural events. For this reason, academia and industry have increased research in the process safety area in the last decade, strengthening quantitative risk analysis (QRA) methodologies for Natech events. However, these methodologies have some gaps that must be closed for a better decision-making process. In this communication a comparative analysis of the existing Natech QRA approaches is done, to highlight features and differences and to identify main gaps that should be addressed in future research. It can be mentioned that all the analyzed methodologies, which have been applied to floods, earthquakes, and lightning, are based on an initial one developed in 2007. The critical gap is that in all these methodologies, the final step is the risk calculation based on fatalities, and they do not consider the area-wide as an essential element in the risk analysis process.

**Keywords:** Natech; process safety; natural event; risk; area-wide

## 1. Introduction

Hazards for people and the environment can emerge when the technological and natural worlds meet each other, and Natech events are testimony of that. Experience also teaches that any kind of natural hazard can trigger Natech events, and a significant natural event is not necessarily required to provoke the release of hazardous materials [1]. Even though these events have originated severe technological accidents, for example, the Kocaeli earthquake (Turkey, 1999) that seriously damaged diverse industrial facilities and an oil refinery, or the Tohoku earthquake-tsunami (Japan, 2011) leading to a serious nuclear accident in the Fukushima power plant, Natech accidents have been often caused by rain, lightning, landslides, or freeze, among others. For instance, in 1994 a lightning impact on three storage tanks containing 15,000 tons of oil in Egypt caused a massive fire; due to the fire, there were more than 400 deaths, oil spills, and water contamination.

Some works have been addressed to know the state of the art of Natech-related research. Nascimento and Alencar [2] carried out a systematic review of research on Natech events between 2000 and 2015, showing that quantitative analyses have had a higher development than qualitative and data analysis; furthermore, those analyzed quantitative methodologies apply mainly to floods, followed by earthquakes and seismic events. Cruz and Suarez-Paba [3], and Suarez-Paba et al. [4] carried out a qualitative meta-analysis of Natech research, analyzing the quantitative and qualitative approaches have been developed to address Natech risk management; more specifically, it was found that quantitative methodologies have been designed for earthquakes, while events like landslides and

extreme temperatures have been rather scarcely studied as a leading cause of Natechs. Caputo et al. [5] made an exhaustive study of the existing methodologies for the quantitative risk analysis of the seismic impact on chemical process plants, reaching the conclusion that more complex risk assessment methodologies should be developed; Caputo et al. [6] developed as well a methodology to estimate the resilience of process plants, applying it to the specific case of earthquake impact Finally, Mesa-Gómez et al. [7] presented a state of the art about Natech qualitative, semi-quantitative, and quantitative analysis methodologies for single and multi-hazard approaches. In this work, the authors highlight that quantitative risk analysis has been widely studied, although its application has been limited to floods, earthquakes, and lightning. All in all, Natech QRA has been improved over time, but its application is still limited to some natural events.

In different documents, it has been highlighted that the scope of current Natech risk analysis should include area-wide risk assessment [4,8,9]. Nonetheless, few methodologies include surrounding areas in their analysis. In the first instance, a qualitative methodology to assess the Natech risk in urban areas was proposed [10]. This methodology considers domino effects, consequences for people, and damage to lifelines in the surrounding area of an industrial plant; the principal limitation is that due to its qualitative approach, a detailed Natech risk analysis is needed. Antonioni et al. [11] presented an application of quantitative risk assessment, with a domino effect approach, to an industrial park. These authors achieved to incorporate domino scenarios in a methodology to evaluate their consequences in the area-wide using human vulnerability models; the central gap is that the methodology does not take into account natural events as hazards, i.e., Natech events are not included. Suda [12] suggested and applied a combined area-wide quantitative risk assessment methodology for earthquakes, but the risk calculation is based only on fatalities. Finally, using the steps done by Suda [12], Kabir et al. [13] developed a Natech risk assessment related to earthquake using a Bayesian belief network model; it considers the physical effects of the resulting chemical accident on people located in the surrounding area, and the risk calculation is also based on fatalities. Thus, even though there have been several efforts to include area-wide in Natech analysis scopes, additional research is still required to close that gap because the risk calculation does not consider environmental and economic consequences.

Considering the findings of advances and scope of Natech QRAs, this work presents a comparison between the different current Natech quantitative risk analyses to highlight features and differences and to identify the main gaps that should be addressed in future research. The main questions that are raised are: What are the differences between current Natech QRAs? Which new elements should be considered in Natech QRAs?

## 2. Current Natech Quantitative Risk Analysis Methodologies

To answer the raised questions, seven methodologies about Natech risk analysis with a quantitative approach were identified. The majority of those methodologies were found in the literature reviews on Natech events carried out by Nascimento and Alencar [2], Cruz and Suarez-Paba [3], Suarez-Paba et al. [4], and Mesa-Gómez et al. [7]. Table 1 presents the Natech QRAs considered in this work.

**Table 1.** Available Natech QRAs.

| Document | Year | Source |
|---|---|---|
| A methodology for the quantitative risk assessment of major accidents triggered by seismic events | 2007 | Antonioni et al. [14] |
| Development of a framework for the risk assessment of Natech accidental events | 2009 | Antonioni et al. [15] |
| Quantitative assessment of domino and Natech scenarios in complex industrial areas | 2014 | Cozzani et al. [16] |
| Quantitative assessment of risk due to Natech scenarios caused by floods | 2015 | Antonioni et al. [17] |
| Quantitative assessment of risk due to major accident triggered by lightning | 2016 | Necci et al. [18] |

| Texas LPG fire: Domino effects triggered by natural hazards | 2018 | Naderpour and Khakzad, [19] |
| Quantitative risk assessment of domino effect in Natech scenarios triggered by lightning | 2020 | Misuri et al. [20] |

## 3. Analysis and Comparison of Methodologies

### 3.1. Natech Quantitative Risk Analyses: Advances and Gaps

The objective of a Natech QRA is the risk calculation, so the frequency of events and their final consequences must be estimated appropriately. The beginning of those QRAs was the study of the intensity and frequency of occurrence of some natural events and their effects on equipment (mainly in atmospheric storage tanks). One of the first studies developed to analyze the impact of earthquakes on storage tanks was published in 2003 [21], the authors analyzed the intensity and likelihood of incidence of earthquakes and therefore the vulnerability of atmospheric storage tanks subjected to seismic actions. Following this research, Fabbrocino et al. [22] integrated the structural seismic risk into a quantitative probabilistic risk analysis through a study case. Those researches gave a technical basis to develop the methodologies shown in Table 1. To contextualize, a summary of each methodology is presented below.

Antonioni et al. [14] proposed a methodology for the individual and societal risk calculation in Natech events triggered by seismic events. The analyzed equipment were vertical and horizontal storage tanks. Additional work was developed by Campedel et al. [23], who strengthened the use of vulnerability models for a group of equipment categories and allowed the assessment of different simultaneous scenarios that may be triggered by the impact of the external event not only on storage tanks, but also on industrial systems where hazardous materials are present (e.g., vertical and horizontal vessels, pipes, pipelines, and others).

Antonioni et al. [15] proposed a framework for the individual and societal risk calculation to Natech events triggered for whatever natural event based on the previous methodology [14]; nevertheless, the frequency, damage models, and case studies were specific to earthquakes and floods. The analyzed equipment was horizontal pressurized storage tanks, anchored and unanchored atmospheric tanks. This methodology was implemented in a computational tool based on Geographic Information System (GIS) and risk re-composition called ARIPAR-GIS (Bologna, Italy).

Analyzing the approaches for the study of domino effects and the Natech framework proposed by Antonioni et al. [15], Cozzani et al. [16] noticed that several analogies were present, so these authors proposed two methodologies to evaluate Natech accidents and domino effects in industrial parks, respectively. The main contribution of the research was the application of a QRA taking into account second-level domino effect scenarios. Two study cases were carried out: the first one was a Natech accident for seismic and flood events, and the other one was a domino effect analysis without considering any natural event as a hazard. The analyzed equipment were pressurized and atmospheric storage tanks.

Antonioni et al. [17] proposed a methodology for the individual and societal risk calculation in Natech events triggered by floods. The difference between this methodology and the one proposed in previous work [15] is the implementation of new equipment vulnerability models. The analyzed equipment were pressurized and atmospheric vessels.

Necci et al. [18] proposed a methodology for the individual and societal risk calculation in Natech events triggered by lightning that impact atmospheric storage tanks. In this research, the authors presented models for the damage probability calculation.

Naderpour and Khakzad [19] presented a risk assessment methodology that considers the potential Natech domino effects and uses Bayesian network capabilities. This work could be applied to all Natech events, but the case study was the Valero refinery (Texas, U.S., 2007) accident originated by water freezing inside a pipe at a deasphalting unit.

Complementing the approach developed by Necci et al. [18], Misuri et al. [20] presented a quantitative methodology for the risk assessment due to domino effects caused by Natech accidents triggered by lightning. The authors incorporated probabilistic models for domino escalation based on the Probit approach and combinatorial analysis by a general procedure that assesses lightning risk. The analyzed equipment were once more atmospheric and pressurized storage tanks.

Figure 1 presents the first [14] and last [20] methodologies developed. As can be seen, although 13 years passed between these two publications, both methodologies have the same main steps: preliminary data gathering, assessment of primary scenarios, and risk calculation.

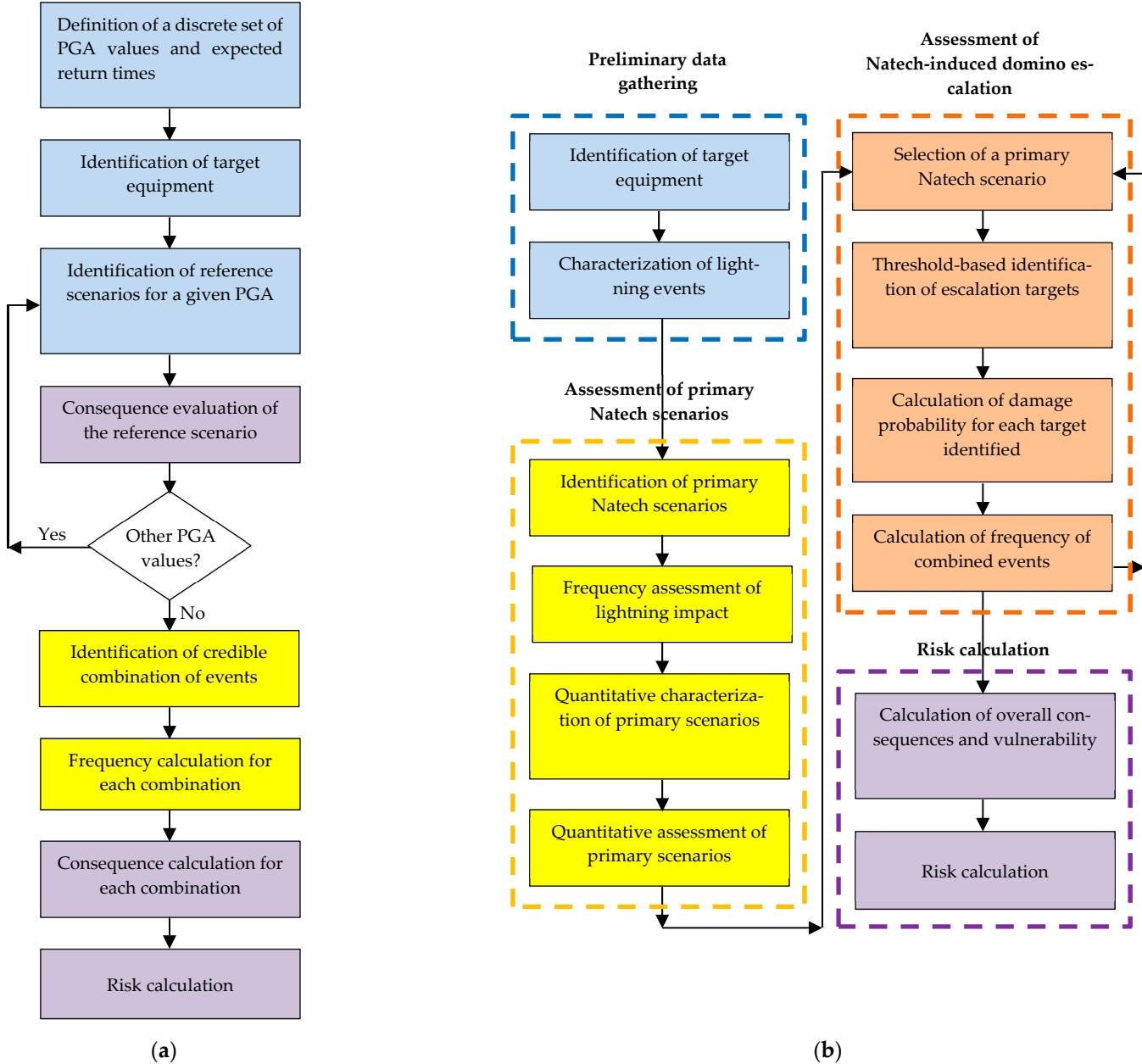

**Figure 1.** (**a**) Flowchart of the first Natech QRA proposed (modified from Antonioni et al. [14]) (reprinted from J. Hazard. Mater, 147, Antonioni, G., Spadoni, G., Cozzani, V. A methodology for the quantitative risk assessment of major accidents triggered by seismic events, 48-59, Copyright (2007), with permission from Elsevier.). (**b**) Flowchart of the last Natech QRA proposed (modified from Misuri et al. [20]) (reprinted from J. Loss Prev. Process Ind, 64, Misuri, A., Antonioni, G., Cozzani, V. Quantitative risk assessment of domino effect in Natech scenarios triggered by lightning, 104095, Copyright (2020), with permission from Elsevier).

A significant contribution that has improved the Natech risk analysis process is the inclusion of the domino effect into the general methodology, for some natural events. This contribution is essential because, in the majority of the cases, domino scenarios have a high probability of occurrence when a Natech happens [16]. A detailed comparison of the seven aforementioned Natech QRA approaches is presented in Table 2.

**Table 2.** Natech QRAs comparison.

| Source | Natural Event | Identified Steps |
|:---:|:---:|:---:|
| Antonioni et al. [14] | Earthquake | Preliminary data gathering<br>Assessment of primary scenarios<br>Risk calculation |
| Antonioni et al. [15] | Flood and earthquake | Preliminary data gathering<br>Assessment of primary scenarios<br>Risk calculation |
| Cozzani et al. [16] | Flood and earthquake | Preliminary data gathering<br>Assessment of primary scenarios<br>**Domino effect (not integrated with Natech) ***<br>Risk calculation |
| Antonioni et al. [17] | Flood | Preliminary data gathering<br>Assessment of primary scenarios<br>Risk calculation |
| Necci et al. [18] | Lightning | Preliminary data gathering<br>Assessment of primary scenarios<br>Risk calculation |
| Naderpour and Khakzad [19] | General | Preliminary data gathering<br>Assessment of primary scenarios<br>**Domino effect (integrated with Natech) ****<br>Risk calculation |
| Misuri et al. [20] | Lightning | Preliminary data gathering<br>Assessment of primary scenarios<br>**Domino effect (integrated with Natech) ****<br>Risk calculation |

Notes: * Domino effect in a different methodology (without Natech approach) ** Domino effect as a step of the Natech QRA.

All the current Natech QRA methodologies consider three essential steps in their procedure, the domino effect step being considered only in three of them; this is a new topic that should be addressed in any new research. Additionally, although those methodologies could be applied to all natural events, only earthquakes, floods, and lightning have been studied.

3.1.1. Preliminary Data Gathering

The first step of a Natech QRA is the preliminary data gathering, which includes:

- Selection of reference scenarios for natural events of concern
- Characterization of the external event: assessment of the intensity of the impact vector

In this step, the authors of the methodologies proposed an impact vector for each natural event analyzed. For earthquakes, Antonioni et al. [14], Antonioni et al. [15], and Cozzani et al. [16] take the horizontal component of peak ground acceleration (PGA) as the primary variable for characterizing seismic events. Equation (1) was used to measure the intensity of the impact vector:

$$f_i = f(PGA_i) \qquad (1)$$

where *PGA<sub>i</sub>* is the ith element of the PGA vector, representing a PGA value.

For the case of floods, the impact vector has been defined by two key parameters: flood water speed and floor water depth [15–17]. For the case studies presented in each research, those parameters were borrowed from hydrogeological studies of the analyzed zone or were given by competent local authorities.

Finally, for lightning, Misuri et al. [20] and Necci et al. [18] used the flash density at ground level as vector impact. This density may be obtained from literature sources. If that parameter is not available, these authors proposed to use a statistical correlation for calculating it:

$$n_g = 0.024 \times T_d^{1.29} \tag{2}$$

where $n_g$ is the flash density at ground level expressed in flashes/(km²·year), and $T_d$ is the yearly number of hours of thunderstorms recorded at the site.

Thus, in the preliminary data gathering step, the Natech QRAs analyzed consider the same models to characterize the natural events, depending on the external event being studied. At this initial point, the methodologies do not present differences.

3.1.2. Assessment of Primary Natech Scenarios

The second step of a Natech QRA is the assessment of primary Natech scenarios, which include:

- Identification of damage types and reference scenarios
- Estimation of the failure and damage frequency/probability
- Identification of credible combination of events
- Frequency/probability calculation for each combination

At this step, the identification of damage types and reference scenarios depend on the type of equipment being analyzed. It was found that atmospheric storage tanks were widely evaluated, followed by pressurized storage tanks and pressurized vessels (vertical and horizontal). The variation between the Natech QRAs approaches is in the estimation of the failure frequency/probability. For earthquakes, Antonioni et al. [14], Antonioni et al. [15], and Cozzani et al. [16] used fragility curves to calculate the failure probability for some equipment instead of vulnerability models based on structural analysis. The following equation was used to measure the intensity of the impact vector and generate fragility curves (vulnerability):

$$Y = k_1 + k_2 Ln(PGA) \tag{3}$$

where *PGA* is the horizontal component of peak ground acceleration and the constants $k_1$, $k_2$ can be found in the literature.

For floods, Antonioni et al. [15] argued that there are limited data available in literature sources to analyze the damage to industrial equipment due to floods, so these authors developed a damage model, relating maximum ranges of water height and water speed to different equipment damage probabilities. In this manner, damage probability values could be estimated. Later on, Cozzani et al. [16] used a simplified approach developed by Landucci et al. [24] to estimate the failure probability for atmospheric tanks storing liquids affected by flood events. Finally, Antonioni et al. [17] extended the use of the simplified approach [24] to develop the fragility model for pressurized and atmospheric horizontal vessels. Those approaches could be improved with the fragility assessment of chemical storage tanks developed by Khakzad and Van Gelder [25], who implemented logistic regression to obtain fragility functions for different failure modes such as floatation, buckling, and sliding. The authors claim that the methodology can be applied to fragility evaluation of different process vessels, although it has been demonstrated only on atmospheric storage tanks.

For lightning, both Necci et al. [18] and Misuri et al. [20] calculated the probability of failure using a model based on a statistical distribution of lightning energy that allows

calculating the probability of perforation in storage tanks; the following equation was used:

$$Ln(P_{p,j}) = 0.924 - 0.908t \tag{4}$$

where $P_{p,j}$ is the probability of perforation for the j-th item, and t is the shell thickness in mm.

All in all, in the assessment of the first Natech scenarios step, the estimation of failure probability is a complex process, so each Natech QRA methodology uses different models and methods to estimate this variable. This could be a possible reason that explains why the application of these methodologies has been limited to three natural events: Earthquakes, floods, and lightning.

3.1.3. Assessment of Natech-Induced Domino Escalation

The third step of a Natech QRA is the assessment of Natech-induced domino escalation, which include:

- Selection of a primary Natech scenario
- Threshold-based identification of escalation targets
- Calculation of damage probability for each target identified
- Calculation of frequency of combined events

This step is an important contribution to Natech QRAs and, even though a method of probability prediction of domino effect triggered by lightning in chemical tank farm was proposed by Yang et al. [26], the specific step about domino effect as part of a Natech QRA is included in the methodologies proposed by Naderpour and Khakzad [19] for whatever natural event and Misuri et al. [20] for lightning.

In a general way, Naderpour and Khakzad [19] estimated the probability at different levels of domino effect using a Bayesian network. The authors calculated the probability of the domino effect ($P_D$) as the multiplication of the probability of the primary event ($P_P$) and the conditional escalation probabilities of the impacted units ($P_E$):

$$P_D = P_P \times P_E \tag{5}$$

At the first level of the domino effect, it is assumed that the primary effect (E) damages at least one of the close units or pipelines, i.e., *U2* or *SP2*. Therefore, the authors represented the probability of the first-level domino effect as:

$$P_{L1} = P(E) \times P(U2 \cup SP2|E) \tag{6}$$

The domino effects propagate to the second level when at least a tertiary unit or pipeline i.e., *U3* or *SP3*, is affected by first-level domino accidents. Therefore, the probability of the second-level domino effect is as follows:

$$P_{L2} = P(E) \times P(U2 \cup SP2|E) \times P(U3 \cup SP3|E, U2, SP2) \tag{7}$$

For lightning, Misuri et al. [20] argued that the main factor in this step is the calculation of each overall escalation scenario. Equation (8) presents the model of calculation:

$$P_E^{(k,m)} = \prod_{l=1}^{n} \left[ 1 - P_{D,l} + \delta(l, J_m^k)(2P_{D,l} - 1) \right] \tag{8}$$

where $P_E^{(k,m)}$ is the probability of the escalation scenario involving *k* targets units simultaneously, $J_m^k$ is a scenario identification vector, whose elements $\gamma_j(j = 1;...; k)$ are the indexes of the *k* secondary events that take place during the overall escalation scenario, m indicates that the overall escalation scenario is the *m-th* (*m = 1;…;$N_k$*) combination of k secondary events, $\delta(l, J_m^k)$ is equal to 1 if the *l-th* secondary event belongs to the vector $J_m^k$, 0 if not, and $P_{D,l}$ is the failure probability of the *l-th* target, which is calculated into the methodology.

The frequency of the *m-th* overall escalation scenario originated from a single primary Natech scenario and simultaneously involving $k$ secondary targets $f_E^{(k,m)}$, may be calculated with the following expression:

$$f_E^{(k,m)} = f_{P,Natech} P_E^{(k,m)} \tag{9}$$

where $f_{P,Natech}$ is the frequency of the first Natech scenario.

Based on the results of past accident analyses, primary Natech scenarios are considered mutually independent, and simultaneous first events are not considered [18]. It should be noted that, as reported in the literature [16], this procedure can be recursively extended to further level domino scenarios.

In conclusion, the methodology proposed by Misuri et al. [20] improves the Natech QRA for lightning, since a step about the domino effect is included. Additional efforts are required to extend the scope of the other methodologies adding the domino effect step in their procedures.

### 3.1.4. Risk Calculation

The final step of a Natech QRA is the risk calculation, which includes:

- Vulnerability/consequences evaluation and calculation by scenario
- Calculation of overall vulnerability and consequences
- Individual and social risk calculation

The final output of all the Natech QRAs approaches is essentially the same: Individual and societal risk. These risk values can be used by the administration to allow or not the activity, and help industrial plant owners to make decisions about actions to reduce risk levels. However, they are based on the number of fatalities and, usually, the analyses do not include the consequences on area-wide. Area-wide refers to the surrounding area of an industrial plant or an industrial park that is vulnerable to natural events and Natech events and can undergo their effects and consequences not only regarding fatalities, but also environmental and economic consequences. Efforts are required to include area-wide in Natech QRAs [4,8,9] because Natech area-wide scenarios could support prevention strategies that adjust to the reality of industrial facilities.

### 3.2. *Analysis Including the Area-Wide Approach*

Three methodologies have been found which include the area-wide scenarios. The first one is a Preliminary qualitative Assessment of Natech Risk in urban areas (PANR) proposed by Cruz and Okada [10]. It considers not only the risk from domino effects, but also the vulnerability in the surrounding area due to the demographic characteristics and the risk for essential facilities from the natural hazard. The authors used the expression defined by Covello and Merkhoher [27], who argued that the risk depends on the hazard and the vulnerability of the elements subjected to the hazard as:

$$Risk = Hazard \times Vulnerability \tag{10}$$

The PANR methodology is composed of several steps that are: data collection and inventory development, hazard identification and vulnerability analysis, and estimation of a Natech Risk Index (NRI) for each hazardous material contained in storage tanks [10]. With the estimation of the NRI values, it is possible to identify the areas with high Natech risk in order to define risk reduction measures. This methodology can be used as a planning tool to determine the effect of risk reduction measures. Thus, once these measures are implemented, new NRI values must be estimated.

Using the expression for risk (Equation (10)), an NRI for each stored hazardous material *i* in a territory exposed to a given natural disaster scenario can be defined as:

$$[NRI_i] = [HRL_i] \times ([D_i] + [Area\_sc_i] + [C_i]) \tag{11}$$

where $HRL_i$ is a score that indicates the hazardous substances release likelihood of each hazardous material $i$ contained in a storage tank given the natural disaster scenario. $D_i$ is a score that measures the potential domino effects or consequences of the hazardous material released from the storage tank $i$ on other storage tanks containing also hazardous materials and located within the directly affected area given the natural disaster scenario. $Area\_sc_i$ is a score that measures the potential consequences of the hazardous materials release from each storage tank $i$ on the population over the directly affected area given the natural disaster scenario. $C_i$ is a score that measures the potential consequences of the release from each hazardous material containing storage tank $i$ on essential facilities located within the directly affected area, being critical for the safety and well-being of the community and the environment [10].

In Equation (11), $HRL_i$, $D_i$, $Area\_sc_i$, and $C_i$ are dimensionless variables; so $NRI_i$ is also a dimensionless variable, whose value is a score that qualitative estimates the risk on an urban area subjected to the spill of a hazardous material contained in a storage tank due to an earthquake. This score can take values ranging between 3 and 75, depending on the qualitative qualification of the independent variables, which can take values between 1 and 5. For instance, the score of $HRL_i$ can be 1 if there is a low likelihood of release or can be 5 if there is a high likelihood of release. For the case of $D_i$, the score is 1 if the domino effect is very low to no potential, and the score is 5 if the domino effect is very high.

The limitation of PANR methodology is that the calculated NRIs do not provide details regarding the expected consequences in the surrounding area because NRI is calculated just qualitatively. In order to estimate the consequences, for example, the number of fatalities in that area, an additional quantitative risk assessment method is required. Thus, a methodology that combines both area-wide and quantitative risk assessment is needed.

Another methodology was proposed by Suda [12]. This work considers models for toxic dispersion and explosions scenarios. It has five steps for the Natech QRA, and this QRA is combined with an evacuation analysis. The framework is presented in Figure 2. This framework includes a scope delimitation to narrow down the area of interest that would be subjected to the effects of Natech events, and its application is for Natechs triggered by earthquakes.

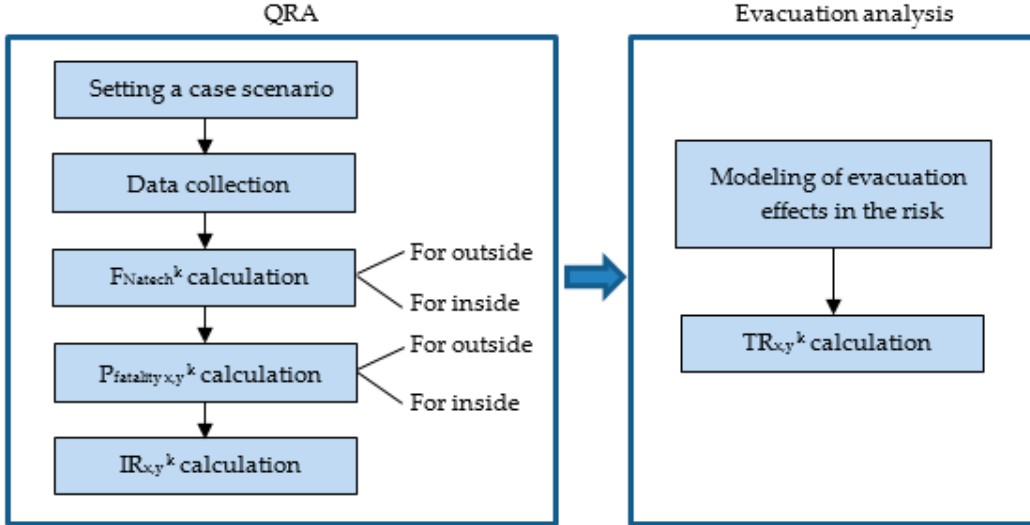

**Figure 2.** Framework for area-wide QRA (modified from Suda [12]).

Step 1: Setting a case scenario. This step requires narrowing down the scope to identify and delimit the area that would be subjected to potential Natech events. The scenario should consider both natural and technological threats.

Step 2: Data collection. This step entails the collection of information related to natural hazards, the industrial facility, building stock, the environment, and other information.

Step 3: Calculation of the Natech frequency. The Natech event frequency occurring at unit k ($F_{Natech}^k$) is calculated by natural hazard parameters and models (e.g., the onsite PGA resulting from the maximum earthquake magnitude expected in a certain return period). The loss of containment (LOC) of hazardous material at the industrial facility is the main aspect of this step.

Step 4: Estimation of fatalities. This step deals with the estimation of the physical effects and consequences on people close to the industrial plant when a technological accident occurs. $P_{fatality x,y}^k$, which is the probability of dying of an individual in the area in the case of a Natech occurring at the equipment/unit *k*.

Step 5: Calculation and visualization of individual risk: individual risk was calculated by numerical simulations for each point (*x,y*) using Python (Netherlands) built to Arcmap (Redlands, California, U.S.).

Once all the data needed are collected, the numerical calculation starts. First, the frequency of a Natech involving the storage tank *k*, $F_{Natech}^k$, will be calculated by using the collected data. This step focuses on just the consequences of the natural hazard event on industrial plants. The analysis will proceed to calculate the physical effects on the surrounding area due to the Natech event, and the corresponding $P_{fatality x,y}^k$ calculation, probability of dying of an individual in area *i* if a Natech occurs at the storage tank *k*. $P_{fatality x,y}^k$ depends on the characteristics and properties of the substances stored in the storage tank *k* (for explosion scenarios and toxic dispersion), and the distance x between the individual and the tank. The QRA last step will be the calculation of the $IR_{x,y}^k$ by using $F_{Natech}^k$ and $P_{fatality x,y}^k$ with the following expression:

$$IR_{x,y}^k = F_{Natech}^k \times P_{fatality x,y}^k \tag{12}$$

where $IR_{x,y}^k$ is the risk to an individual at the point (*x,y*) from a storage tank *k*. $F_{Natech}^k$ is the frequency of a Natech involving the storage tank *k*, and $P_{fatality x,y}^k$ is the probability of death of an individual at the point (*x,y*) for the same accident. This equation is the basic expression of the quantitative risk assessment in this framework. $IR_{x,y}^k$ depends of course on the location of the individuals.

In the evacuation analysis, this research defines Total Risk (*TR*). $TR_{x,y}^k$ is the total risk to an individual during his/her evacuation from the point (*x,y*) to a safer point in the event of the Natech at storage tank *k*. As an individual evacuates his/her location changes, hence $IR_{x,y}^k$ changes. The authors defined $TR_{x,y}^k$ as the weighted sum of $IR_{x,y}^k$ that an individual experiences during his/her evacuation. $TR_{x,y}^k$ is calculated as:

$$TR_{x(0),y(0)}^k = \sum_{t=0}^{t=T} IR_{x(t),y(t)}^k \tag{13}$$

Finally, Kabir et al. [13] presented a knowledge-based Bayesian belief network (BBN) model for risk assessment scenarios triggered by earthquakes under a quantitative approach, considering only the explosion scenario. In this model, the authors used the five-step approach following Suda [12], which are presented in Figure 2 for QRA.

To calculate the frequency of primary LOC, ($F_{Natech}^k$), the authors used the software RAPID-N [28]. This software is an open and collaborative web-based application allowing the user to estimate the probability of the release of hazardous materials by using earthquake parameters.

To estimate the physical effects, the authors used BREEZE Haz Pro, an air dispersion model for accidental chemical releases. $P_{fatality x,y}^k$ largely considers properties of the substance stored in the equipment/unit k, and it depends on the distance between the hazard and the individual. The calculation of individual risk ($IR_{x,y}^k$) associated with the explosion of a flammable cloud is the final step of the risk assessment. This risk is calculated as the product of $F_{Natech}^k$ and $P_{fatality x,y}^k$, as expressed in Equation (12). Additionally, in this methodology, the probability of occurrence of a fatality and the occurrence of the Natech event are considered not conditional. However, the probability of the occurrence of the

fatality is a function of the hazardous materials release and overpressure associated with the damage of the tank.

The consequences were calculated combining both an explosion model and a vulnerability model. The authors considered the explosion due to the release, dispersion, and ignition of flammable substances. The algorithm includes models to estimate the consequences of blast pressure on structures and humans ($P_{fatalityx,y}^k$). The results of the explosion scenario were mapped using ArcMapVR and ArcGISVR (Redlands, California, U.S.).

The BBN model for the Natech risk assessment was implemented in Netica software. For this model, six parent or independent nodes, and 16 child or dependent nodes are connected with 25 links. For the development of the model, 4975 conditional probabilities are considered. Netica uses a junction tree or join tree algorithm for the final Bayesian network inference.

The limitation of the methodologies proposed by Suda [12] and Kabir et al. [13] is that the risk calculation is based only on fatalities. It means that although the integration of a Natech QRA with an evacuation analysis helps decision-makers in the design of prevention and emergency strategies to assist people, environmental and economic effects have to be additionally calculated to expand the scope of the QRA considering whole consequences in the area-wide. Furthermore, all the methodologies that consider area-wide were applied to earthquakes or seismic events case studies. Nevertheless, those methodologies could be modified to apply to other Natech events (e.g., triggered by floods, winds, tsunamis, and so on). All in all, a methodology that combines all consequences for area-wide and quantitative risk assessment is needed for different Natechs.

## 4. Future Research Needs

The analysis and comparison of the current Natech QRA methodologies show that the efforts to improve risk analysis in industrial plants subjected to the effects of natural events have increased with time. The implementation of these QRAs has been essentially focused on particular Natech events, specifically earthquakes, tsunamis, and floods; besides the intensity that these events can have, another reason for this is probably the strong consequences on the population that they can have (cases of Kocaeli or Tohoku, for example) and their considerable impact on the media. Nevertheless, other natural events, often less spectacular, such as lightning, rain, or freezing, can also cause Natech accidents.

An additional effort should be done to study these cases and to develop specific methodologies or to adapt the existing ones to allow a quantitative analysis of their risk. Once more, the estimation of failure frequency/probability is a difficult process due to the complexity of some natural events and to the lack of information about some of them, and more research is required to get more information in order to assess the primary Natech scenarios.

Domino effect in industrial plants and, more specifically, in chemical process plants, has been intensively studied in the last decades, as well as its integration in the quantitative risk analysis methodologies; and domino effect as a component of Natech accidents has also been studied, but with much less intensity. A research effort would be also required to fill this gap.

Another field in which rather few contributions have been made in Natech risk analysis is the area-wide approach. This approach has key elements to improve prevention and mitigation plans in areas close to industrial parks, where people, flora and fauna, and infrastructures are present. It has been highlighted that a complete Natech risk assessment is needed not only to help industrial plant owners to make decisions, but also to improve the safety of area-wide industrial parks. From this point of view, the final individual and social risk assessment over the area is required, as well as the potential effects and consequences on the environment (soil, flora, and fauna). The PANR methodology is a significant contribution because it considers the majority of the variables inside and outside of an industrial plant; however, its qualitative approach is not detailed, and the subjectivity is high. On the other hand, although the quantitative area-wide proposed methodologies

base the individual risk calculation on the position of a person, which is useful to improve evacuation and emergency plans, they have been applied only to earthquakes. Thus, further research is required to extend the use of these methodologies to other Natechs.

In general terms, the available Natech QRAs methodologies have the same main steps: preliminary data gathering, assessment of primary scenarios, and risk calculation. In each step, the differences between each methodology are about the variables and models used to characterize the natural event. One additional step has been added in one methodology that evaluates domino effects due to a Natech triggered by lightning, but its application should be extended to other natural events. In the final step, the analyses take into account human consequences (fatalities) by calculating the individual and societal risk. This approach is repeated in the quantitative area-wide proposed methodologies, so it is important to note that new elements should be considered in future research about Natech QRA in order to (1) extend the scope to other natural events, (2) strengthen domino effect analysis, (3) incorporate area-wide approach in the current methodologies taking advantage on their advances, and (4) extend the scope of risk calculation adding environmental, infrastructure, and economic consequences, not only inside the industrial plant, but also outside considering the area-wide.

## 5. Conclusions

The frequency of Natech accidents has increased in the past decades, due to both the increase in the number of existing chemical process and storage plants and the length of installed pipelines transporting hazardous materials. Furthermore, it seems probable that the frequency of certain natural events, such as floods, will increase due to climate change, with the consequent effect on the occurrence of Natech events. Additionally, these accidents are often associated with important damages, not only because of the loss of human lives, but also because of material losses and environmental impact.

An important research effort has been and is being done in the study of Natech accidents in industrialized countries. Interesting and useful methodologies have been developed to analyze them, to predict their effects and consequences and, also, to predict their risk. Nevertheless, several gaps exist in the quantitative assessment of these accidents. Even though the quantitative risk analysis of technological accidents—essentially in the field of chemical and energy industries—has significantly improved in recent years, the quantitative treatment of the domino couple "natural event—technological accident" keeps being complex and involving certain gaps in which additional research is required. In this communication, an attempt has been done to identify them.

**Author Contributions:** Conceptualization, investigation, methodology, formal analysis, writing—original draft preparation: A.M.-G.; writing—review and editing: A.M.-G., J.C., M.S.-S., and F.M. All authors have read and agreed to the published version of the manuscript.

**Funding:** This research was partially funded by the Spanish Ministry of Economy and Competitiveness (project CTQ2017-85990-R, co-financed with FEDER funds). Additionally, the Ministry of Science, Technology, and Innovation of Colombia has contributed with financial resources.

**Informed Consent Statement:** Not applicable.

**Data Availability Statement:** No new data were created or analyzed in this study. Data sharing is not applicable to this article.

**Acknowledgments:** A. Mesa-Gómez thanks to the Spanish Ministry of Economy and Competitiveness for the resources and the Ministry of Science, Technology, and Innovation of Colombia for the Ph.D. scholarship (MINCIENCIAS grant No. 785, 2018-2021).

**Conflicts of Interest:** The authors declare no conflict of interest.

**Nomenclature**

| | |
|---|---|
| $Area\_sc_i$ | Score that measures the potential consequences of the hazardous materials release from each storage tank i on the population |
| $C_i$ | Score that measures the potential consequences of the release from each hazardous material contained in storage tank i on essential facilities |
| $D_i$ | Score that measures the potential domino effects or consequences of the hazardous material released from the storage tank i |
| $E$ | Primary consequence (*CO*) for a domino effect |
| $f_E^{(k,m)}$ | Frequency of the m-th overall escalation scenario originated from a single primary Natech scenario and simultaneously involving k secondary targets |
| $f_i$ | Impact vector for each natural event analyzed |
| $F_{Natech}^k$ | Frequency of a Natech occurring at the storage tank k |
| $f_{P,Natech}$ | Frequency of the primary Natech scenario |
| $HRL_i$ | Score that indicates the Hazardous substances Release Likelihood of each hazardous material i contained in a storage tank |
| $IR_{x,y}^k$ | Risk to an individual at the point (x,y) from a storage tank k |
| $J_m^k$ | Scenario identification vector |
| $k_1$ | Probit constant |
| $k_2$ | Probit constant |
| $n_g$ | Flash density at ground level expressed in flashes/(km$^2$·year) |
| $NRI_i$ | Natech Risk Index |
| $P_D$ | Probability of domino effect |
| $P_{D,l}$ | Failure probability of the l-th target |
| $P_E$ | Probabilities of the impacted units |
| $P_E^{(k,m)}$ | Probability of the escalation scenario involving simultaneously k targets units |
| $P_{fatality\ x,y}^k$ | Probability of dying of an individual at the point (x,y) in the case of a Natech occurring from the storage tank k |
| PGA$_i$ | *i*-th element of the PGA vector, representing a PGA value |
| $P_{L1}$ | Domino effect at the first level |
| $P_{L2}$ | Domino effect at the second level |
| $P_P$ | Probability of the primary event (*PP*) |
| $P_{p,j}$ | Probability of perforation for the j-th item |
| $SP2$ | Pipelines for domino effect analysis at first level |
| $SP3$ | Pipelines for domino effect analysis at the second level |
| $t$ | Tank shell thickness in mm |
| $T_d$ | Yearly number of hours of thunderstorm recorded at the site |
| $TR_{x(0),y(0)}^k$ | Total risk to an individual during his/her evacuation from the point (x,y) to a safer point under the Natech at storage tank k |
| $U2$ | Process unit for domino effect analysis at first level |
| $U3$ | Process unit for domino effect analysis at the second level |
| $Y$ | Intensity of the impact vector |
| $\delta(l, J_m^k)$ | 1 if the *l*-th secondary event belongs to the vector $J_m^k$, 0 if not |

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
