# Peer review of "Advances and Gaps in Natech Quantitative Risk Analysis"

_processes, doi:10.3390/pr9010040_

Round 1

Reviewer 1 Report

This paper try a comparative analysis of the existing Natech Quantitative Risk Analysis approaches highlighting features and differences and trying to identify main gaps that should be addressed in future research.

The main concern is that all the literature on the multi-risk assessment that include cascading effect is not considered. The implication between NaTech and TechNa hazards is not addressed and/or discussed. There are various papers on the quantitative evaluation of the impacts on the industrial plants and the consequent impact on the environment and human factors by extreme weather conditions (extreme wind, extreme sea waves, extreme rainfall, tsunamis, component fatigue,induced seismicity, etc.)

Some aspects are treated very superficially, e.g.
Line 342 I think "incredible" write something like that "maximum credible earthquaKe". I know the maxmimum magnitude expected in an area; the most recurrent magnitude,the chacteristic earthquake and so on.

I think this paper is acceptable after a major revision.

Author Response

Point 1: The main concern is that all the literature on the multi-risk assessment that include cascading effect is not considered. The implication between NaTech and TechNa hazards is not addressed and/or discussed. There are various papers on the quantitative evaluation of the impacts on the industrial plants and the consequent impact on the environment and human factors by extreme weather conditions (extreme wind, extreme sea waves, extreme rainfall, tsunamis, component fatigue, induced seismicity, etc.)

Response 1: This comment was partially addressed. The introduction was strengthened with some references that deal with Natech events and focus more on plant components than fatalities. These references were included from line 48 to line 52. However, Due to the scope of the paper, that is specific for Quantitative Risk Analysis methodologies for Natech events, the analysis is limited to highlight features and differences and to identify main gaps in those methodologies. All the literature on multi-risk assessment that include cascading effect is not considered because many of those resources are not related with Natech QRA, or they were analyzed in review papers with a broader scope, for example:

Suarez-Paba, M.C., Perreur, M., Munoz, F., Cruz, A.M. Systematic literature review and qualitative meta-analysis of Natech research in the past four decades. Saf. Sci. 2019, 116, 58–77. https://doi.org/https://doi.org/10.1016/j.ssci.2019.02.033

Mesa-Gómez, A., Casal, J., Muñoz, F. Risk analysis in Natech events: State of the art. J. Loss Prev. Process Ind. 2020, 64, 104071. https://doi.org/https://doi.org/10.1016/j.jlp.2020.104071

Point 2: Some aspects are treated very superficially, e.g.

Line 342 I think "incredible" write something like that "maximum credible earthquaKe". I know the maxmimum magnitude expected in an area; the most recurrent magnitude, the chacteristic earthquake and so on.

Response 2: The paragraph that contents the expression “maximum credible earthquake” was changed as follow (from line 356 to line 359): “Step 3: Calculation of the Natech frequency. The Natech event frequency occurring at unit k (FNatechk) is calculated by natural hazard parameters and models (e.g., the onsite PGA resulting from the maximum earthquake magnitude expected in a certain return period). The Loss of Containment (LOC) of hazardous material at the industrial facility is the main aspect of this step”

Reviewer 2 Report

In order to identify main gaps that should be addressed in future research, in this paper, the authors present a comparative analysis of existing Natech quantitative risk assessment (QRA) approaches and highlight features and differences. They point out that critical gaps in all these methodologies focus on fatalities calculations and do not consider the area size as an essential element in the risk analysis process.

The paper is interesting due to both relevance and complexity of the problem. Nonetheless, one weakness is present in this paper. It is listed here for the authors’ convenience.

In order to render the paper more interesting and comprehensive, the authors should draw a specific section, where they could  more precisely elaborate the future research generically sketched in the Section Conclusions

With respect to the presentation of the material the following constructive criticism is offered.

Some additional references that deal with NaTech events and focus more on plant components than fatalities could be cited.

Antonio C. Caputo, Fabrizio Paolacci, Oreste S. Bursi, Renato Giannini, Problems and Perspectives in Seismic Quantitative Risk Analysis of Chemical Process Plants, J. Pressure Vessel Technol. February 2019, 141(1): 010901. doi: https://doi.org/10.1115/1.4040804

Antonio C. Caputo, Bledar Kalemi, Fabrizio Paolacci, Daniele Corritore, Computing resilience of process plants under Na-Tech events: Methodology and application to sesmic loading scenarios, Reliability Engineering & System Safety, Volume 195, March 2020,

The Reviewer suggests minor revisions to this paper.

Author Response

Point 1: In order to render the paper more interesting and comprehensive, the authors should draw a specific section, where they could more precisely elaborate the future research generically sketched in the Section Conclusions. 

Response 1: As the reviewer suggested, a new section called “Future research needs” was included in the numeral 4 (line 419). In this section, detailed analysis and explanation about current gaps and the interest of studying and covering them are presented. Additionally, the “5. Conclusions” section was strengthened (from line 463 to line 477).

Point 2: Some additional references that deal with NaTech events and focus more on plant components than fatalities could be cited.

Antonio C. Caputo, Fabrizio Paolacci, Oreste S. Bursi, Renato Giannini, Problems and Perspectives in Seismic Quantitative Risk Analysis of Chemical Process Plants, J. Pressure Vessel Technol. February 2019, 141(1): 010901. doi: https://doi.org/10.1115/1.4040804

Antonio C. Caputo, Bledar Kalemi, Fabrizio Paolacci, Daniele Corritore, Computing resilience of process plants under Na-Tech events: Methodology and application to sesmic loading scenarios, Reliability Engineering & System Safety, Volume 195, March 2020.

Response 2: The references that the reviewer gave us were obtained and checked, and certainly considered interesting and therefore cited in the paper, specifically in the introduction section, from line 48 to line 52. The text and references added are:

“Caputo et al. [5] made an exhaustive study of the existing methodologies for the quantitative risk analysis of the seismic impact on chemical process plants, reaching the conclusion that more complex risk assessment methodologies should be developed; Caputo et al. [6] developed as well a methodology to estimate the resilience of process plants, applying it to the specific case of earthquake impact”

  1. Caputo, A. C., Paolacci, F., Bursi, O. S., Giannini, R. Problems and perspectives in seismic quantitative risk analysis of chemical process plants. J. Pressure Vessel Technol. 2019, 141, 010901-15. https://doi.org/10.1115/1.4040804

6.         Caputo, A. C., Kalemi, B., Paolacci, F., Corritore, D. Computing resilience of process plants under Na-Tech events: Methodology and application to seismic loading scenarios. Reliab. Eng. Syst. Saf. 2020, 19, 106685. https://doi.org/10.1016/j.ress.2019.106685
